# Electrochemotherapy of Melanoma Cutaneous Metastases in Organ Transplant Recipients: A Systematic Review of Preclinical and Clinical Studies

**DOI:** 10.3390/ijms24098335

**Published:** 2023-05-06

**Authors:** Sara Milicevic, Maja Cemazar, Andreja Klevisar Ivancic, Gorana Gasljevic, Masa Bosnjak, Gregor Sersa, Barbara Peric

**Affiliations:** 1Institute of Oncology Ljubljana, Zaloska Cesta 2, 1000 Ljubljana, Slovenia; smilicevic@onko-i.si (S.M.); mcemazar@onko-i.si (M.C.); aklevisar@onko-i.si (A.K.I.); ggasljevic@onko-i.si (G.G.); mbosnjak@onko-i.si (M.B.); gsersa@onko-i.si (G.S.); 2Faculty of Medicine, University of Ljubljana, Zaloska 2, 1000 Ljubljana, Slovenia; 3Faculty of Health Sciences, University of Primorska, Polje 42, 6310 Izola, Slovenia; 4Faculty of Pharmacy, University of Ljubljana, Askerceva Cesta 7, 1000 Ljubljana, Slovenia; 5Faculty of Health Sciences, University of Ljubljana, Zdravstvena Pot 5, 1000 Ljubljana, Slovenia

**Keywords:** cutaneous melanoma, electrochemotherapy, immune checkpoint inhibitors, tumor microenvironment, immunosuppression

## Abstract

Cutaneous melanoma is a highly aggressive form of skin cancer. The development of immune checkpoint inhibitors (ICIs) has revolutionized the management of advanced melanoma, led to durable responses, and improved overall survival. However, the success of ICIs in melanoma treatment is influenced by the tumor microenvironment (TME) which plays a critical role in regulating the immune response to the tumor. Understanding the mechanisms underlying this interaction is crucial to optimizing the efficiency of ICIs. Electrochemotherapy (ECT) has been shown to enhance the efficacy of ICIs in melanoma treatment by inducing tumor cell death and facilitating the release of tumor antigens which can subsequently be recognized and targeted by the immune system. Moreover, ECT has been reported to modulate the TME, leading to increased infiltration of immune cells and a more favorable immunological profile. In this review, we summarize the available knowledge of changes in TME after ECT of melanoma cutaneous metastasis and highlight the differences in tumor-infiltrating immune cells between immunocompetent and immunosuppressed organisms. In addition, we showed that ECT can be an effective and safe procedure for organ transplant recipients. Furthermore, repeated ECT may enhance immune activation and probably induce a bystander effect by trained immunity.

## 1. Introduction

Malignant melanoma is one of the immunogenic tumors with a high potential to elicit a specific adaptive antitumor immune response [1]. Since the introduction of v-raf murine sarcoma viral oncogene homolog B1 (BRAF) and methyl ethyl ketone protein kinase (MEK) inhibitors, followed by immunotherapy with immune checkpoint inhibitors (ICIs), the overall survival of patients with stage III and IV melanoma has greatly improved [2]. Despite these advances in melanoma treatment, a large subset of melanoma patients shows primary or secondary resistance [3]. To improve the clinical response to ICIs, novel therapeutic strategies that combine local therapies with ICIs are under investigation [4,5,6,7].

Electrochemotherapy (ECT) is a very safe and efficient local anticancer treatment frequently used in treating in-transit melanoma metastases and displays immunostimulatory properties through immunogenic cell death [8]. A comprehensive analysis of the International Network for Sharing Practices of ECT (InspECT) database has shown an 82% per-tumor overall response (OR) and a 64% complete response (CR) rate in patients with melanoma after a single application ECT [9,10]. Recently, a retrospective study showed that patients treated with pembrolizumab and electrochemotherapy (ECT) experienced lower disease progression rates and longer survival than those who received pembrolizumab alone, suggesting that ECT might improve patient outcomes by acting as an in situ vaccination, boosting the ICI response [11].

Tumor-infiltrating immune cells participating in the control of malignant cells with the extracellular matrix and tumor vasculature, which together represent the tumor microenvironment (TME), continue to be areas of intense research. Innate and adaptive immune responses are both responsible for the immune surveillance of melanoma [12]. High rates of helper T (Th) and cytotoxic T lymphocytes (CTL), natural killer cells (NK), dendritic cells (DCs), and proinflammatory M1 macrophages are associated with improved overall survival, while high levels of regulatory T cells (Tregs), myeloid-derived suppressor cells, and anti-inflammatory M2 macrophages promote tumor progression [1]. Not only the local TME, but also the responsiveness of peripheral immune cells may shift the balance toward either immune elimination or immune evasion of cancer cells [13]. In more detail, tumors with an inflamed phenotype, characterized by high T-cell infiltration, increased interferon gamma (IFN-γ) signaling, expression of programmed death-ligand 1 (PD-L1), and high tumor mutational burden, tend to be more responsive to ICIs [14].

The most efficient combination of local therapeutic options and systemic therapies is still unknown. Although tremendous effort has been made to explore the TME, it still presents challenges of high heterogenicity, complexity, and plasticity. To address this challenge, we integrate the available knowledge of changes in TME (especially immune cell infiltration) after ECT of melanoma cutaneous metastasis, with special emphasis on differences in tumor-infiltrating immune cells between immunocompetent and immunosuppressed organisms. Considering that less attention has been devoted to changes in TME and clinical results after ECT in organ transplant recipients on immunosuppressant medication, we present a case of advanced melanoma in which repeated treatment with ECT with bleomycin-induced a complete clinical response of in-transit metastases.

## 2. Response of TME to Electrochemotherapy

The host immune system is considered to be an essential part of the response to ECT. In 1997, Sersa et al., observed both a twofold longer tumor growth delay and higher curability rates in immunocompetent C57B1/6 mice than in immunodeficient nude mice when mice were treated with ECT with cisplatin [15]. Similarly, a study investigating the difference in the antitumor effectiveness of ECT with cisplatin combined with intramuscular mIL-12 gene electrotransfer in murine mammary adenocarcinoma in immunocompetent BALB/c and immunodeficient SCID mice (C. B-17/IcrHanHsd-Prkdcscid) resulted in cell killing of 1.5 versus 1.9 log. In immunocompetent mice, one dose of mIL-12 resulted in 22% of tumor cures and prolongation of a specific tumor growth delay, while in SCID mice, even multiple intramuscular mIL-12 gene electrotransfer had no effect on specific tumor growth delay [16]. Although these two early studies showed a worse response to ECT treatment in immunosuppressed organisms, the detailed observation of changes in TME was not the point of the studies.

In a study on murine melanoma, markers of immunogenic cell death were studied after ECT with cisplatin and oxaliplatin. Calreticulin (CRT) exposure was observed after ECT with both drugs used in vitro and in vivo. The release of CRT shifts nonimmunogenic cell death to immunogenic cell death attracts antigen-presenting cells (APCs) and consequently induces an adaptive immune response. As a result, after ECT, a 4-fold increase in tumor infiltration by granzyme B-positive (GrB) immune cells, representing mainly NK cells and CTLs, were observed. This finding demonstrated that exposure of melanoma cells to ECT with cisplatin or oxaliplatin induces immunogenic cell death which correlates with an increase in tumor-infiltrating lymphocytes (TILs) after ECT [17]. In addition, the exposure of cells to electric pulses leads to reactive oxygen species (ROS) formation which was also confirmed in melanoma cells in vitro [18,19]. ROS signaling cascades play a role in the propagation and execution of cell death further stimulating the innate immune system [20].

CRT surface expression was also detected in human melanoma metastases after ECT with bleomycin. Moreover, immediate and late changes in T-cell infiltration in ECT-treated lesions using fluorescent immunohistochemistry were also investigated. Ten patients (median age 80.5 years) who had not been eligible for systemic therapies have had histologically confirmed cutaneous or subcutaneous melanoma metastases located in the limb and have been undergoing ECT with bleomycin were included. Skin biopsies were performed on different lesions prior to ECT and one and fourteen days after ECT. Before ECT, Th cells were the most represented T cells in the perilesional border, while Tregs were present in both the perilesional dermis and within the tumor. After ECT, Tregs significantly decreased in number, while CTLs significantly increased in the perilesional dermis 14 days after ECT. In contrast, the frequency of Th remained constant before and after ECT [21].

Another immunohistochemical analysis of the inflammatory infiltrate in sequential punch biopsies (before ECT and ten min, three h, three days, ten days, one month, and two months after ECT) of human melanoma cutaneous metastases after ETC showed similar results. In this study, only two patients were enrolled. To minimize differences, two sets of biopsies were obtained in one patient, one set from a single large nodule and another set from smaller lesions. Unfortunately, the authors did not report if any differences in these two sets of biopsies were observed. However, they described differences in histologic findings as well as in immune infiltration in all time lapses after ECT. Early presence and a progressive increase in inflammatory infiltrate were observed with T lymphocyte prevalence in all samples taken at different time points after ECT. From the earliest phases after ECT, CTLs increased in number and surrounded the tumor cells, and residual CTL infiltration was observed two months after ECT. Research has demonstrated that CTLs are activated early after ECT and are key elements in the late inflammatory response [22]. In contrast, Ths were rare and unaltered as previously described [21,22]. Sparse NK cells were observed 3 h after ECT and were still detected one month after treatment, while no CD20+ B cells were detected [22].

To observe DC activation and migration, sequential biopsies before ECT with bleomycin and seven and fourteen days after ECT in nine patients with histologically confirmed skin melanoma metastasis were performed. Prior to ECT, the most represented subgroup of DCs was Langerhans cells (LCs), located mainly in the epidermis and expressing a typical immature immunophenotype. Epidermal DCs significantly decreased seven days after ECT, but LCs repopulated fourteen days later, suggesting that ECT may stimulate LCs to drain lymph nodes [23]. These findings are in line with a previous murine study, although in the latter, the changes were seen earlier after ECT treatment [24]. Moreover, following ECT, DCs significantly increased in number and positivity for the CD83 marker, indicating maturation [23].

## 3. Case Report of ECT

### 3.1. ECT Procedure

A 68-year Caucasian woman, 18 years after liver transplantation, and taking the calcineurin inhibitor tacrolimus (Prograf^®^) underwent excision of pT3b melanoma from the right lower limb in March 2019. Due to a positive sentinel node and after FDG PET CT excluded visceral involvement, groin dissection was performed. In total, 3 out of 23 nodes were positive. After one week of receiving targeted therapy, it was discontinued due to side effects. Immunotherapy was contraindicated due to organ transplantation. After approximately one year, in-transit melanoma metastases were cytologically confirmed. During a six-month period, three ECT sessions with i.v. bleomycin application wasperformed, and one ECT session was performed after nine months on only a small number of nodules.

The procedures were performed under general anesthesia. All four ECT treatments were performed according to a standardized protocol, described in the publication Updated standard operating procedures for electrochemotherapy of cutaneous tumors and skin metastases, published in 2018 [25].

Electric pulses were generated by a Cliniporator (IGEA S.P.A., Carpi, Italy) device and applied using either plate or hexagonal electrodes, depending on the size of the tumor lesion. Bleomycin (Bleomedac, Medac, Wedel, Germany), at a standard dose of 15,000 IU/m^2^ body surface area, at the first ECT procedure, and at a reduced dose of 10,000 IU/m^2^ at all other procedures was administered intravenously in a 60 s time frame. Between 8 and 28 min after bleomycin infusion, electrodes were placed on the nodules, including a 1 cm safety margin and electric pulses were applied thereafter. For plate electrodes eight pulses with voltage 960 V (for 8 mm electrodes), pulse duration 100 μs and frequency 5 kHz were applied and for hexagonal electrodes 12 pulses with voltage 730 V, pulse duration 100 μs, and frequency 5 kHz were applied. The amplitudes were automatically checked after each application. If needed, several applications were performed to cover the entire nodule volume. Table 1 shows the characteristics of each ECT session.

Regression of treated lesions was observed while new metastases also appeared (Figure 1C). FDG PET CT in March 2021 showed clinically silent skeletal metastases in the 7th thoracic vertebra, 5th lumbar vertebra, left iliac bon,e and manubrium of the sternum. In November 2021, the fourth ECT of in-transit metastases was performed. Clinically, complete response was observed (Figure 1F), and five months later, only one vital CM nodule was present which was excised under local anesthesia (using 2% Xylocaine). At the same time, we also excised two pretreated nodules with clinically complete response after ECT. On regular follow-up, nine months after the last ECT, no vital in-transit melanoma metastasis was observed. No serious adverse effects were observed after ECT, and only mild local pain, swelling, and mild scarring in the treated area were noticed. In September 2022, liver metastases were discovered on total body CT scan. The patient died in October 2022.

### 3.2. Histologic and IHC Findings

All specimens were fixed in 10% buffered formalin overnight. Histological analysis of morphology and estimation of tumor-infiltrating lymphocytes (TILs) were performed on 2–3 μm thick, formalin-fixed paraffin-embedded (FFPE) sections stained with hematoxylin-eosin (HE).

Immunohistochemical (IHC) characterization of the tumor inflammatory infiltrate was performed on 2–4 µm FFPE tissue sections dried at 56 °C for 2 h using the fully automated IHC staining platform Benchmark Ultra (Manufacturer Ventana ROCHE Inc., Tucson, AZ, USA). Epitopes were retrieved onboard employing heat-mediated epitope retrieval using high pH Cell Conditioning Solution one (cat. No. 950-124; manufacturer Ventana ROCHE Inc., Tucson, AZ, USA) for 88 min at 100 °C. Retrieved epitopes were detected using commercially available monoclonal primary antibodies, as presented in Table 2. All antibodies were diluted using DAKO REAL™ antibody diluent (cat. No. S2022; manufacturer DAKO Agilent, Santa Clara, CA, USA) and incubated on board for 60 min at 37 °C. Specifically, bound primary antibodies were visualized using a three-step multimer detection system OptiView DAB IHC Detection Kit (cat. No. 760-700; Ventana ROCHE Inc., Tucson, AZ, USA) according to the manufacturer’s instructions.

The amount of TIL was estimated semi-quantitatively according to the MIA scoring [26] independently by two experienced pathologists. Lymphocyte and macrophage distribution densities were estimated semi-quantitatively according to Park CH et al. [27]. Briefly, the lymphocyte distribution density was defined as follows: 0 = absence of lymphocytes within the tissue, 1 = presence of lymphocytes occupying <25% of the tissue, 2 = presence of lymphocytes occupying 25 to 50% of the tissue, and 3 = presence of lymphocytes occupying >50% of tissue. Additionally, the percentage of tumor tissue covered by lymphocytes/macrophages was estimated. Evaluation of positive cells was performed at 40× in at least five high-power fields with high lymphocyte/macrophage density. In regard to CD4 and CD56 staining, only small round cells with strong positivity were scored as lymphocytes since CD4 also stains macrophages and CD56 weakly stains a smaller portion of melanoma cells (differentiation was possible on the basis of the morphology and intensity of the positive reaction). The spatial distribution of CD8+ T lymphocytes was additionally determined according to three categories: immune deserted, excluded, and inflamed [28]. The results are shown in Table 3.

Histologic examination of vital in-transit CM nodules excised five months after the fourth ECT showed metastasis of CM in the dermis with expansion into the subcutaneous fat (Figure 2A). The percentage of TILs determined on HE sections was grade I according to the MIA scale [26]. Immunohistochemically, approximately 10% of CD3, 10–15% of CD8, 2% of CD4, 1% of CD56, 1% of FoxP3, and 10–15% of CD163-positive cells were detected intratumorally, and all results fit into grade one according to the Park H et al., criteria [27] (Figure 2C–H). According to the spatial distribution of CTLs, the tumor nodule was defined as inflamed.

Two biopsies of sites of prior cutaneous in-transit lesions treated by ECT showed only fibrosis and dermal melanosis (Figure 3)—complete pathologic response.

## 4. Discussion

In this study, we showed that ECT can be an effective and safe procedure in organ transplant recipients. Repeated ECT may enhance immune activation and probably induce a bystander effect by trained immunity. Furthermore, in viable excised metastases, the number of CD163-positive macrophages was prevalent, an observation that might contribute to ECT resistance.

To our knowledge, no studies describing the TME in immunocompromised melanoma patients after ECT are available. However, histological and immunohistochemical immune characteristics of the TME of mainly primary cutaneous melanoma and some melanoma metastasis biopsy specimens in 20 renal transplant recipients who developed post-transplantation melanoma were reported. Testing adaptive immunity, little or no expression of CD8+, FoxP3, PD1, and PD-L1 was observed, confirming a deserted TME and the absence of T-cell activation [29]. Mechanisms underlying immune desert phenotypes are as follows: lack of tumor antigens, defects in tumor antigen processing and presentation, and dysfunction of DC-T-cell interactions [14].

In contrast to a prior study describing the immune characteristics of the TME in renal transplant recipients, Quaglino et al., investigated the distribution and phenotype of TILs of in-transit metastases before ECT in 15 stage III melanoma patients. A high number of Th cells and CTLs within the TILs was observed in nine patients (60.0%), while more than 10% of Tregs were found in five patients (33.3%). Furthermore, brisk distribution TILs with a high number of CTLs were associated with higher response rates to ECT and a lower incidence of visceral progression [30].

Comparison of the results of previous studies with immunohistochemical characteristics of excised vital in-transit lesions after the fourth ECT in our patient should be conducted with caution. Based on the spatial distribution of CTLs, the excised vital nodule could be described as immune inflamed. In addition, TIL phenotyping with IHC staining for CD3, CD4, CD8, and FoxP3 showed a favorable CD8:FoxP3 ratio. As previously described, these features correlate with improved CTL function and better clinical response rates [28,30,31]. Despite the beneficial spatial distribution of CTLs and the phenotype of TILs, the amount of TILs was grade I according to the MIA scale. This could be one of the reasons that this vital nodule escaped immune control. However, failed immune responses contribute to the selection of the most resilient cancer cell clones that are less likely to undergo immune destruction [13].

In our case, the patient was taking the immunosuppressant tacrolimus which is a calcineurin inhibitor that blocks the activation of the nuclear factor of activated T cells (NFAT) and leads to a decreased synthesis of IL-2. IL-2 is a key cytokine required for T-cell activation of the primary immune response and throughout memory cells as well as T-cell proliferation [32,33]. Calcineurin inhibitors also have an inhibitory effect on mitogen-activated protein kinase (MAPK), resulting in reduced production of IL-2, IL-10, tumor necrosis factor alpha (TNF α), and IFN-γ [34]. Although tacrolimus has a limited effect on the MAPK pathway and functional NFAT signaling occurs within some myeloid cell subsets, it was proven that cells of the innate system are less susceptible to the immunosuppressive effect of tacrolimus than cells of the adaptive immune system [33,35].

The innate immune system is less vulnerable to the immunosuppressive effect of tacrolimus, so it is not surprising that we found 10–15% positivity of CD 163 (macrophage- and monocyte-specific transmembrane protein) in excised vital in-transit metastasis [36]. This marker is representative of anti-inflammatory M2 macrophages and infiltration of this type of macrophage in this vital nodule could be another reason for its resistance to ECT-triggered immunomodulatory effects. Namely, it was previously shown that expression of CD 163 is induced by anti-inflammatory IL-10 as well as glucocorticoids and IL-6, whereas inflammatory stimuli, such as lipopolysaccharide, TNF α, and IFN-γ, lead to a rapid downregulation of expression [36,37]. In addition, CD163 macrophages play a central role in immunosuppression and are associated with a poor prognosis in different cancers, including melanoma. In contrast, specific depletion of CD163 macrophages results in tumor regression and massive infiltration of activated T cells, including both CD4+ and CD8+ cells [36].

Interestingly, histological examination of sites previously treated for in-transit CM metastasis by ECT showed only fibrosis and dermal melanosis, confirming the clinically observed complete response after ECT. However, after the first two ECT procedures, out-of-field progression of a high number of in-transit metastases was observed, while after the third ECT, only six in-transit metastases appeared, and after the fourth ECT, only one in-transit metastasis escaped of immune control. These findings suggest that repeating ECT potentiates antitumor immune effects as previously reported [38,39]. Our results imply that repeated ECT boosts the antitumor response locally and enhances immune activation, probably by additional exposure to tumor-associated antigens. Moreover, less impairment of the innate immune system by tacrolimus and evidence from previous preclinical studies support that the observed bystander effect could also be induced by trained innate immunity, as tacrolimus has a major immunosuppressive effect by acting on T cells and does not damage the innate immune system defense capacity [8,40,41].

### Study Limitations

Tumor response to ECT is very complex and depends on many factors. Response to ECT is dependent mainly on tumor size, type, and previous oncological treatment. Therefore, the comparison of different studies with their inherent risk of selection bias is challenging. Due to intermetastatic heterogeneity, significant variations among different metastatic lesions in the same patient as well as the primary tumor, including the composition of the TME, can be observed [13,42]. We also acknowledge that interpreting immunohistochemical studies using different methodologies and various antibodies is a limitation of our review. In addition, we have focused mainly on tumor immune infiltrate, while other parts, such as stroma and tumor vasculature, which also play important roles, were not in our point of observation.

## 5. Conclusions

Our case report with a review of the literature contributes to the understanding of changes in the melanoma tumor microenvironment, especially immune cell infiltration, even in the case of immunosuppression. This shows that ECT is a safe and effective procedure for organ transplant recipients. Repeated ECT may enhance immune activation and probably induce a bystander effect by trained immunity.

ECT could be utilized to enhance the activation of melanoma-specific immunity and maintain the long-term antimelanoma immune response as part of a combined therapeutic approach in the era of ICI treatment.

## Figures and Tables

**Figure 1 ijms-24-08335-f001:**
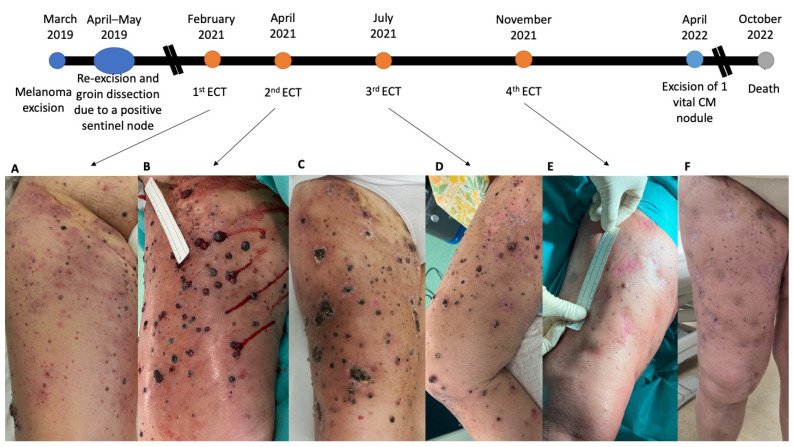
Treatment course of in-transit melanoma metastases; (**A**) 1st ECT; (**B**) 2nd ECT; (**C**) 1 month after the 2nd ECT; (**D**) 3rd ECT; (**E**) 4th ECT; (**F**) 1 month after the 4th ECT.

**Figure 2 ijms-24-08335-f002:**
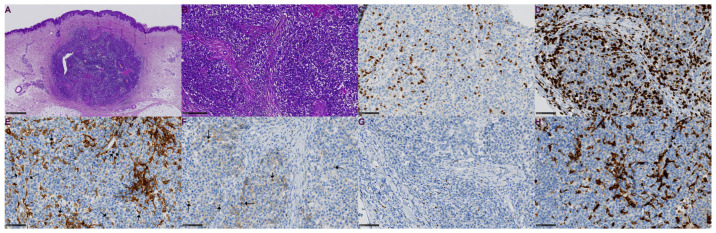
Histopathology of excised vital in-transit CM nodule 5 months after 4th ECT; (**A**) H&E, 2×; (**B**) H & E, 10×; (**C**) CD3, 20×; (**D**) CD8, 20×; (**E**) CD4, 20×; lymphocytes indicated by arrows; (**F**) CD56, 20×; lymphocytes indicated by arrows; (**G**) FoxP3, 20×; (**H**) CD163, 20×.

**Figure 3 ijms-24-08335-f003:**
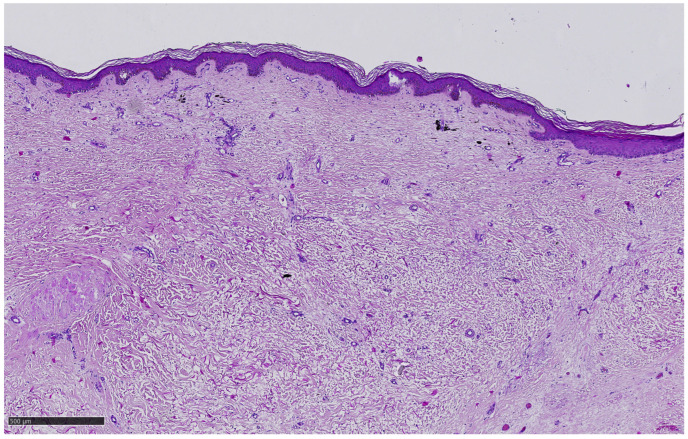
Histopathology of excised completely regressed pretreated in-transit CM nodule 5 months after the 4th ECT, H&E staining, 2×.

**Table 1 ijms-24-08335-t001:** Characteristics of each ECT session.

ECT Session	Time of ECT (No. of Months after 1st Session)	Number of Nodules	Size of Nodules (Diameter)	No. of Pulses	Type of Electrodes
1st	0	31 smaller + 2 larger	<8 mm and >17 mm	2 per smaller nodule + 13 per larger nodule	Plate + hexagonal
2nd	2 months	47	Up to 15 mm	47	hexagonal
3rd	5 months	30	Up to 20 mm	70	hexagonal
4th	9 months	6	Up to 12 mm	25	hexagonal

**Table 2 ijms-24-08335-t002:** Antibodies used for IHC characterization.

Antibody	Type of Monoclonal Antibody	Clone	Cat. No.	Manufacturer	Diluted at
CD4	rabbit	SP35	104R	CellMarque, Rocklin, CA, USA	1:10
CD8	mouse	C8/144B	M7103	DAKO Agilent, Santa Clara, CA, USA	1:100
CD56	rabbit	MRQ-42	156R	CellMarque, Rocklin, CA, USA	1:200
CD163	mouse	MRQ-26	163M	CellMarque, Rocklin, CA, USA	1:200
FOXP3	rabbit	EP340	AC-0304RUO	Epitomics, Burlingame, CA, USA	1:200

**Table 3 ijms-24-08335-t003:** Results of immunostaining scores and determination of spatial distribution of CD8+ T lymphocytes independently evaluated by two experienced pathologists.

	Pathologist 1	Pathologist 2
TIL	Grade 1	Grade 1
CD3	1 (10%)	1 (10%)
CD4	1 (2%)	1 (2%)
CD8	1 (15%)	1 (10%)
CD56	1 (1%)	1 (1%)
FoxP3	1 (1%)	1 (1%)
CD163	1 (10%)	1 (15%)
Spatial distribution of CD8	inflammed	inflammed

## Data Availability

The data presented in this study are available on request from the corresponding author.

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
