# Peer review of "Electrochemotherapy of Melanoma Cutaneous Metastases in Organ Transplant Recipients: A Systematic Review of Preclinical and Clinical Studies"

_ijms, 2023, doi:10.3390/ijms24098335_

Round 1
Reviewer 1 Report
Overall, the research is interesting to the reader; however, I found that the manuscript is suitable for a case report, so I am not sure if it belongs in a review article. I suggest the author arrange the way to present the data as a case report; the rest could be added to the discussion.
Examples
1. https://casesjournal.biomedcentral.com/articles/10.1186/1757-1626-1-212
2. https://www.ncbi.nlm.nih.gov/pmc/articles/PMC3437114/
3. https://onlinelibrary.wiley.com/doi/abs/10.1111/cup.13829?
4. https://www.sciencedirect.com/science/article/abs/pii/0190962291700465
Additional minor,
Please use the full name before you abbreviated it ex. MEK and BRAF
Reviewer 2 Report
The authors of the review titled “Electrochemotherapy of melanoma cutaneous metastases in organ transplant recipients: a systematic review of preclinical and clinical studies” have presented a comprehensive review stating the importance of resident immune cells in the tumor microenvironment of cutaneous melanoma in the context of electrochemotherapy. The authors have cited examples of several previous studies where chemotherapeutic drugs like pembrolizumab has been used in combination with ECT. There are few major comments which should be addressed:
1. The first paragraph of Section 2 is unnecessary and can be eliminated.
2. In the lines 97 to 105, authors have mentioned the effect of Calreticulin exposure. Is there any literature on other stress markers like ROS or NFK-β?
3. Please double check if in Table 2, the CD4 antibody dilution is 1:10 or 1:100?
4. In Figure 2, there are 2 Gs in the caption
5. For Figure 2, is there any before ECT treatment image? Same goes for Figure 3. This should be placed if available.
English language is readable and easily understandable.
Round 2
Reviewer 1 Report
I could accept the authors' explanation, and the author has also addressed my minor concern. Thus, I recommend that the manuscript be published in the IJMS journal.
Reviewer 2 Report
Comments have been satisfactorily addressed.